# COVID-Net CXR-S: Deep Convolutional Neural Network for Severity Assessment of COVID-19 Cases from Chest X-ray Images

**DOI:** 10.3390/diagnostics12010025

**Published:** 2021-12-23

**Authors:** Hossein Aboutalebi, Maya Pavlova, Mohammad Javad Shafiee, Ali Sabri, Amer Alaref, Alexander Wong

**Affiliations:** 1Department of Computer Science, University of Waterloo, Waterloo, ON N2L 3G1, Canada; 2Waterloo Artificial Intelligence Institute, University of Waterloo, Waterloo, ON N2L 3G1, Canada; mjshafiee@uwaterloo.ca (M.J.S.); a28wong@uwaterloo.ca (A.W.); 3Department of Systems Design Engineering, University of Waterloo, Waterloo, ON N2L 3G1, Canada; mspavlova@uwaterloo.ca; 4DarwinAI Corp., Waterloo, ON N2V 1K4, Canada; 5Department of Radiology, Niagara Health, McMaster University, St. Catharines, ON L2S 0A9, Canada; Sabri.ali@gmail.com; 6Department of Diagnostic Radiology, Thunder Bay Regional Health Sciences Centre, Thunder Bay, ON P7B 6V4, Canada; Alarefa@tbh.net; 7Department of Diagnostic Imaging, Northern Ontario School of Medicine, Thunder Bay, ON P7B 5E1, Canada

**Keywords:** computer vision, COVID-19, deep neural networks, severity assessment

## Abstract

The world is still struggling in controlling and containing the spread of the COVID-19 pandemic caused by the SARS-CoV-2 virus. The medical conditions associated with SARS-CoV-2 infections have resulted in a surge in the number of patients at clinics and hospitals, leading to a significantly increased strain on healthcare resources. As such, an important part of managing and handling patients with SARS-CoV-2 infections within the clinical workflow is severity assessment, which is often conducted with the use of chest X-ray (CXR) images. In this work, we introduce COVID-Net CXR-S, a convolutional neural network for predicting the airspace severity of a SARS-CoV-2 positive patient based on a CXR image of the patient’s chest. More specifically, we leveraged transfer learning to transfer representational knowledge gained from over 16,000 CXR images from a multinational cohort of over 15,000 SARS-CoV-2 positive and negative patient cases into a custom network architecture for severity assessment. Experimental results using the RSNA RICORD dataset showed that the proposed COVID-Net CXR-S has potential to be a powerful tool for computer-aided severity assessment of CXR images of COVID-19 positive patients. Furthermore, radiologist validation on select cases by two board-certified radiologists with over 10 and 19 years of experience, respectively, showed consistency between radiologist interpretation and critical factors leveraged by COVID-Net CXR-S for severity assessment. While not a production-ready solution, the ultimate goal for the open source release of COVID-Net CXR-S is to act as a catalyst for clinical scientists, machine learning researchers, as well as citizen scientists to develop innovative new clinical decision support solutions for helping clinicians around the world manage the continuing pandemic.

## 1. Introduction

The impact of the COVID-19 pandemic on the health and economy has been unprecedented. While more than one year has been passed since the declaration of the global pandemic by the World Health Organization [1], countries are still struggling with controlling the spread of the SARS-CoV-2 virus that is causing the pandemic. In this regard, the global healthcare system has suffered a devastating impact from this pandemic, with hospitals and clinics overwhelmed by the surge of patients, and thus not all patients can have access to intensive care units for further treatment and care [2]. Furthermore, there have been shortages in PPEs, ventilators, and other medical supplies due to the increasing demand on healthcare resources [2,3].

This significant strain on healthcare resources caused by the COVID-19 pandemic from both a personnel and supplies perspective necessitates improved clinical decision support tools for aiding clinicians and front-line healthcare workers with more efficient and effective clinical resource allocation. A critical part of clinical resource allocation during this pandemic has been severity assessment, which is often conducted with the assistance of chest X-ray (CXR) images. More specifically, radiology indicators within the patient’s lungs such as ground-glass opacities can provide critical information for determining whether the condition of a SARS-CoV-2 positive patients warrants advanced care such as ICU admission and ventilator administration.

Given that severity assessment using CXR images can be quite challenging even for expert radiologists, providing computer-aided clinical support for this task can greatly benefit the hospitals in determining patients’ conditions and respond more quickly to those who may require more advanced treatments or intensive care. While much of research literature have focused on computer-aided COVID-19 detection from CXR images [4,5] and computed tomography (CT) scans [6,7,8,9,10], the area of computer-aided severity assessment is significantly less well explored. Some of the notable works in this area are COVID-Net S, a tailored deep convolutional neural network proposed by Wong et al. [11] to predict the extent scores from CXR images, as well as the study by Cohen et al. [12] where the same extent scores are predicted.

Motivated to extend upon this area in the direction of airspace disease grading, we introduce COVID-Net CXR-S, a convolutional neural network for predicting the airspace severity of a SARS-CoV-2 positive patient based on a CXR image of the patient’s chest, as part of the COVID-Net open source initiative [4,7,8,11]. The paper is organized as follows. Section 2 describes the methodology behind the design and construction of the proposed COVID-Net CXR-S, a demographic and protocol analysis of the dataset used, as well as radiologist validation. Section 3 presents and discusses the quantitative and qualitative results obtained from the experiments evaluating the efficacy of the proposed COVID-Net CXR-S.

## 2. Material and Methods

In this work, we introduce COVID-Net CXR-S, a convolutional neural network tailored for the prediction of airspace severity of a SARS-CoV-2 positive patient based on chest X-ray images. To train COVID-Net CXR-S, we transferred representational knowledge from CXR images of a multinational cohort, and then leveraged the CXR data grouped based on airspace severity levels. We further validated the behavior of COVID-Net CXR-S in a transparent and responsible manner via explainability-driven performance validation, as well as conducted radiologist validation on select cases by two expert board-certified radiologists. The details between network design, data preparation, explainability-driven performance validation, and radiologist validation are described below.

### 2.1. Network Design

The proposed COVID-Net CXR-S architecture is depicted in Figure 1, with additional details on each layer within the architecture detailed in Table 1. More specifically, we leveraged a machine-driven design exploration strategy [13] to construct a backbone architecture tailored for a strong balance between accuracy and efficiency [14]. The constructed backbone architecture exhibits light-weight macro-architecture and micro-architecture designs comprised primarily of depthwise and pointwise convolutions with selective long-range connectivity. We then leveraged transfer learning to transfer representational knowledge gained from over 16,000 CXR images from a multinational cohort of over 15,000 SARS-CoV-2 positive and negative patient cases [15,16,17,18,19,20] via the constructed backbone architecture into a custom network architecture for severity assessment, where a combination of a dense layer and a severity prediction layer is used to predict between two levels of airspace severity. Specifically, 16,352 CXR images with 2358 SARS-CoV-2 positive images and 1505 positive patient cases were leveraged in the transfer learning. The multi-national patient cohort curated by the Radiological Society of North America (RSNA) RICORD initiative [20] was used afterwards for the evaluation of the efficacy of the proposed network.

The COVID-Net CXR-S network and associated scripts are available in an open source manner at http://www.covid-net.ml.

### 2.2. Data Preparation

In this study, we leveraged the RSNA RICORD (International COVID-19 Open Radiology Database) dataset [20] for the severity scoring to train COVID-Net CXR-S after transfer learning. The RSNA RICORD dataset was curated by the Radiological Society of North America (RSNA) as part of a global initiative to assemble an international task force of scientists and radiologists to create a multi-institutional, multinational, expert-annotated COVID-19 imaging dataset. More specifically, we leveraged the airspace disease grading provided by RSNA RICORD, where each lung is split into three separate zones (for a total of six zones) and opacity is studied for each zone. We group the patient cases into two airspace severity level groups: (1) Level 1: opacities in 1–2 lung zones, and (2) Level 2: opacities in 3 or more lung zones. Example CXR images for the different airspace severity level groups from the dataset are shown in Figure 2. This severity level designation was chosen given clinical similarities between patient cases within each airspace severity level groups in terms of the treatment regimen, and thus facilitates clearer guidelines for course of action.

Given this airspace severity level grouping scheme, the RSNA RICORD dataset used in this study consists of 909 CXR images from 258 patients, with 227 images from 129 patients in Level 1 and 682 images from 184 patients in Level 2. We used 150 randomly selected CXR images for the test set, ensuring no patient overlap between the test and train datasets. Table 2 summarizes the demographic variables and imaging protocol variables of the CXR data in the RSNA RICORD dataset used in this study. It can be observed that the patient cases in the cohort used in the study are distributed across the different age groups, with the mean age being 59.11 and the highest number of patients in the cohort are between the ages of 50–69.

The data preparation scripts are available in an open source manner at http://www.covid-net.ml.

### 2.3. Network Training

Training on the COVID-Net CXR-S architecture after knowledge transfer is conducted on the training portion of the aforementioned dataset using the Adam optimizer with a learning rate of 0.0001 for 30 epochs with a batch size of 32. During the training, we account for the imbalance in the number of patient cases between the airspace severity levels by performing batch balancing, where at each epoch we randomly sampled an equal number of CXR images from each severity level for each batch of data. As a pre-processing step, the CXR images were cropped (top 8% of the image) prior to the training process to better mitigate the influence of commonly-found embedded textual information, and resampled to 480 × 480 for training purposes. In addition, we leveraged data augmentation during the training process with the following augmentation types: translation (±10% in x and y directions), rotation (±$10^{\circ }$), horizontal flip, zoom (±15%), and intensity shift (±10%).

The scripts for the aforementioned process are available in an open source manner at http://www.covid-net.ml.

### 2.4. Explainability-Driven Performance Validation

To study the decision-making behavior of the proposed COVID-Net CXR-S network in a transparent and responsible manner, we conducted explainability-driven performance validation using GSInquire [21], which was shown to provide state-of-the-art explanations. More specifically, GSInquire leverages the concept of generative synthesis [13] from the machine-driven design exploration process via an inquisitor I within a generator-inquisitor pair $\left\{G,I\right\}$ to generate quantitative interpretations of the decision-making process of COVID-Net CXR-S on a given CXR image. In this case, the generator G in the generator-inquisitor pair is the optimal generator that was leveraged to construct the backbone architecture during the network design process. The details pertaining to GSInquire for explaining the decision-making behavior of deep neural networks on CXR images can be found in [4]. An interesting property of GSInquire that also makes it well-suited for explainability-driven performance validation is that it is capable of producing explanations identifying specific critical factors within an image that quantitatively impacts the decisions made by a deep neural network, thus making it more readily interpretable and more quantitative for validation purposes than the types of relative importance variations visualized by other methods.

This explainability-driven performance validation process enables the identification of anomalies in decision-making behavior or potential erroneous indicators, leading to invalidate decisions or biases, as well as the validation of whether clinically relevant indicators are leveraged.

### 2.5. Radiologist Validation

The results obtained for COVID-Net CXR-S during the explainability-driven performance validation process for selected patient cases are further reviewed and reported on by two board-certified radiologists (A.S. and A.A.). The first radiologist (A.S.) has over 10 years of experience, and the second radiologist (A.A.) has over 19 years of radiology experience.

## 3. Results and Discussion

To evaluate the efficacy of the proposed COVID-Net CXR-S for the purpose of severity assessment of COVID-19 cases from CXR images, we conducted both quantitative performance evaluation as well as qualitative explainability-driven performance validation. The quantitative and qualitative results are presented and discussed below.

### 3.1. Quantitative Results

The overall quantitative performance assessment results of the proposed COVID-Net CXR-S network on the RSNA RICORD dataset can be seen in Table 3. For comparison purposes, quantitative performance assessment was also conducted on the ResNet-50 [22] network architecture as well as on CheXNet [23], a state-of-the-art deep neural network architecture that has been shown to outperform other network architectures for CXR image analysis tasks. The architectural and computational complexity of COVID-Net CXR-S in comparison to CheXNet [23] and ResNet-50 [22] is also shown in Table 4.

A number of observations can be made from the quantitative results. More specifically, it can be observed that the COVID-Net CXR-S network can achieve a high level of accuracy at 92.66%, which is 9.33% and 5.33% higher than that achieved by CheXNet and ResNet-50 network architectures, respectively. The COVID-Net CXR-S network also achieved strong performance in terms of area under the receiver operating characteristic curve (AUC ROC), achieving a 96.09% AUC which is 12.47% and 4.21% higher than the CheXNet and ResNet-50 networks, respectively. The COVID-Net CXR-S network achieved higher accuracy and AUC at a significantly lower computational complexity of ∼57% and ∼69% (at ∼11.1 G FLOPs) in comparison to the CheXNet and ResNet-50 networks, respectively, as well as a ∼63% lower architectural complexity (at ∼8.8M parameters) than the ResNet-50 network, and achieving similar architectural complexity when compared to CheXNet. Achieving high architectural efficiency and computational efficiency is important for operation in resource-limited clinical scenarios where low-cost computing devices are desired.

It can also be observed that the COVID-Net CXR-S network achieved a sensitivity of 92.3% on the Level 2 cases and sensitivity of 92.85% on the Level 1 cases. Furthermore, it can also be seen that the proposed COVID-Net CXR-S network can achieve high positive predictive value (PPV) of 95.78% and 87.27% for Level 2 cases and Level 1 cases, respectively. This high PPV for Level 2 cases ensures that fewer false positives for more severe cases are reported by COVID-Net CXR-S, which is important since patients with severe conditions require advanced treatment and management and thus high false positive rate can lead to significant burden on the healthcare system where resources are limited. Finally, Table 5 provides a more detailed picture of the performance of COVID-Net CXR-S via the confusion matrix.

Therefore, it can be clearly seen that the COVID-Net CXR-S can successfully predict the severity of COVID-19 infections and distinguish COVID-19 infections from CXR images to make such predictions as evident by the predictive performance shown in Table 3 and the critical factors it leveraged as identified by Figure 3.

### 3.2. Qualitative Results

Results from the conducted explainability-driven performance validation [21] (see Figure 3) show that clinically relevant visual indicators in the lungs were leveraged in the decision-making process. This validation is very important for auditing COVID-Net CXR-S in a transparent and responsible manner to ensure that not only is it leveraging the right indicators for driving the severity assessment process, but also that it is not primarily leveraging erroneous visual indicators (e.g., embedded markers, motion artifacts, imaging artifacts, etc.) to make ’right decisions for the wrong reasons’. Furthermore, this validation has the potential to help in the discovery of additional visual indicators to assist a clinician in their severity assessment as well as improve the trust that clinicians may have during operational use.

### 3.3. Radiologist Analysis

The expert radiologist findings and observations for select patient cases with respect to the identified critical factors during explainability-driven performance validation as shown in Figure 3 are as follows. In all three cases, COVID-Net CXR-S correctly detected them to be patients with Level 2 airspace severity, which were clinically confirmed.

**Case 1.** According to radiologist findings, it was observed by both radiologists that there is patchy airspace opacity in the lower left lung lobe, which is consistent with one of the identified critical factors leveraged by COVID-Net CXR-S;

**Case 2.** According to radiologist findings, it was observed by both radiologists that there are patchy airspace opacities in the left and right midlung regions that coincide with the identified critical factors leveraged by COVID-Net CXR-S in that region. It was further observed by one of the radiologists that there are additional lower lobe opacities in both lungs;

**Case 3.** According to radiologist findings, it was observed by one of the radiologists that there are hilar opacities on the right lung that coincide with the identified critical factors leveraged by COVID-Net CXR-S. It was also observed that there are opacities in the left lower lobe, with the superior aspect of the opacities being leveraged by COVID-Net CXR-S. The second radiologist observed patchy airspace opacities in the right lung that overlap with the critical factors leveraged by COVID-Net CXR-S.

As such, based on the radiologist findings and observations on the three patient cases, it was shown that although some opacities were not identified by GSInquire as critical factors driving the decision-making behavior of COVID-Net CXR-2, several other abnormalities identified as critical factors were consistent with radiologist interpretations. Therefore, based on the critical factors identified by GSInquire as critical factors driving the decision-making behavior of COVID-Net CXR-S, the network was able to differentiate between the airspace severity levels but not necessarily leverage all regions of concern in making its severity assessment decisions.

## 4. Conclusions

In this study, we introduced COVID-Net CXR-S, a convolutional neural network for the prediction of airspace severity of a SARS-CoV-2 positive patient based on a CXR image of the patient’s chest. By leveraging transfer learning, we transferred representational knowledge gained from over 16,000 CXR images from a multinational cohort of over 15,000 SARS-CoV-2 positive and negative patient cases into a custom network architecture for severity assessment. The promising quantitative and qualitative results obtained from the conducted experiments demonstrate that the proposed COVID-Net CXR-S, while not a production-ready solution, can be a potentially become a powerful tool for aiding clinicians and front-line healthcare workers via computer-aided severity assessment of CXR images of COVID-19 positive patients. The ultimate goal for the open source release of COVID-Net CXR-S (http://www.covid-net.ml) is to act as a catalyst for clinical scientists, machine learning researchers, as well as citizen scientists to develop innovative new clinical decision support solutions for helping clinicians around the world manage the continuing pandemic.

## Figures and Tables

**Figure 1 diagnostics-12-00025-f001:**
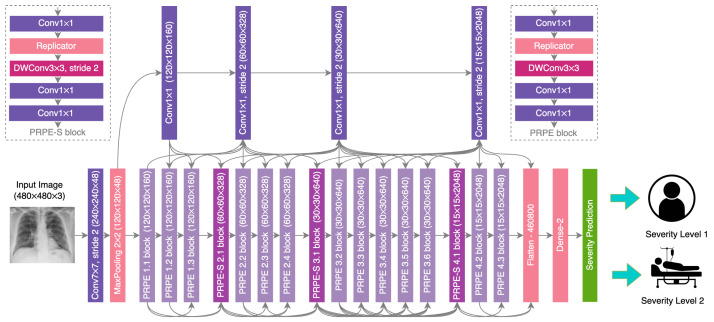
COVID-Net CXR-S network design. The COVID-Net backbone design exhibits high architectural diversity and sparse long-range connectivity, with macroarchitecture and microarchitecture designs tailored specifically for the detection of COVID-19 from chest X-ray images. The network design leverages light-weight design patterns in the form of projection-expansion-projection-expansion (PEPE) patterns to provide enhanced representational capabilities while maintaining low architectural and computational complexities.

**Figure 2 diagnostics-12-00025-f002:**
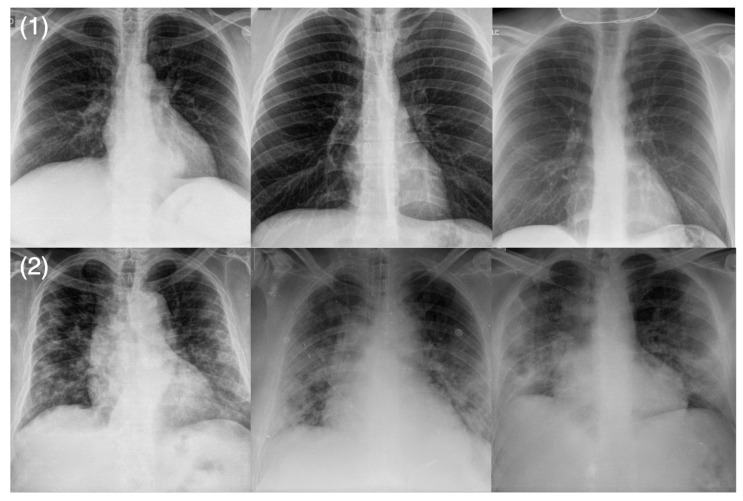
Example chest X-ray images from the RSNA RICORD dataset: (**1**) Level 1 airspace severity: opacities in 1–2 lung zones and (**2**) Level 2 airspace severity: opacities in 3 or more lung zones.

**Figure 3 diagnostics-12-00025-f003:**
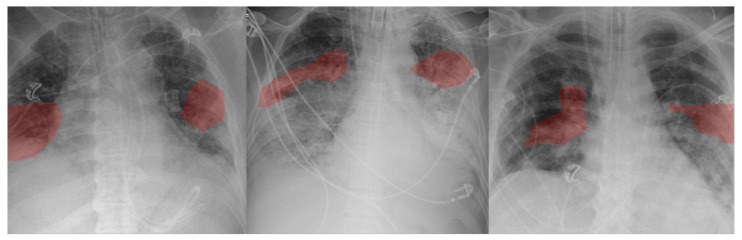
Examples of Level 2 severity patient cases and the associated critical factors (highlighted in red) as identified by GSInquire [21] during explainability-driven performance validation as what drove the decision-making behavior of COVID-Net CXR-S. (**left**) Case 1, (**middle**) Case 2, (**right**) Case 3. Radiologist validation showed that several of the critical factors identified are consistent with radiologist interpretation.

**Table 1 diagnostics-12-00025-t001:** Detailed description of each layer of COVID-Net CXR-S architecture. PRPE refers to layer with convolution layer of filter size 1×1.

Layer Name	Output Size	Specs (Filter Shape, Filter Number)
Conv1	240×240	7×7, 48
PRPE 1.1	120×120	1×1, 160
PRPE 1.2	120×120	1×1, 160
PRPE 1.3	120×120	1×1, 160
PRPE 2.1	60×60	1×1, 328
PRPE 2.2	60×60	1×1, 328
PRPE 2.3	60×60	1×1, 328
PRPE 2.4	60×60	1×1, 328
PRPE 3.1	30×30	1×1, 640
PRPE 3.2	30×30	1×1, 640
PRPE 3.3	30×30	1×1, 640
PRPE 3.4	30×30	1×1, 640
PRPE 3.5	30×30	1×1, 640
PRPE 3.6	30×30	1×1, 640
PRPE 4.1	15×15	1×1, 2048
PRPE 4.2	15×15	1×1, 2048
PRPE 4.3	15×15	1×1, 2048
Dense	2	2048×1, 2

**Table 2 diagnostics-12-00025-t002:** Summary of demographic variables and imaging protocol variables of CXR data in the dataset used in this study. Age and sex statistics are expressed on a patient level, while imaging view statistics are expressed on an image level.

Age	Mean ± Std	59.11±16.19
	<20	2 (0.8%)
	20–29	7 (2.7%)
	30–39	26 (10.1%)
	40–49	37 (14.3%)
	50–59	58 (22.5%)
	60–69	58 (22.5%)
	70–79	42 (16.3%)
	80–89	22 (8.5%)
	90+	6 (2.3%)
**Sex**		
	Male	161 (62.4%)
	Female	97 (37.6%)
**Imaging view**		
	AP	505 (55.6%)
	PA	5 (0.6%)
	Unknown	399 (43.9%)

**Table 3 diagnostics-12-00025-t003:** Sensitivity, positive predictive value (PPV), and accuracy of the proposed COVID-Net CXR-S, CheXNet [23], and ResNet-50 [22]. Best numbers are highlighted in bold.

	Metric	Sensitivity (Level 1)	Sensitivity (Level 2)	PPV (Level 1)	PPV (Level 2)	AUC	Accuracy
Network							
**CheXNet [23]**		**93.88%**	63.46%	82.88%	84.62%	83.62 %	83.33%
**ResNet-50 [22]**		91.84%	78.85%	**89.11%**	83.67%	91.88%	87.33%
**COVID-Net CXR-S**		92.3%	**92.85%**	87.27%	**95.78%**	**96.09%**	**92.66%**

**Table 4 diagnostics-12-00025-t004:** Architectural and computational complexity of the proposed COVID-Net CXR-S, CheXNet [23], and ResNet-50 [22]. Best numbers are highlighted in bold.

Network	Parameters (M)	FLOPs (G)
**CheXNet [23]**	**8.1**	26.0
**ResNet-50 [22]**	23.6	35.5
**COVID-Net CXR-S**	8.8	**11.1**

**Table 5 diagnostics-12-00025-t005:** Confusion Matrix of COVID-Net CXR-S.

Severity Level	Level 1	Level 2
**Level 1**	48	4
**Level 2**	7	91

## Data Availability

All datasets, data preperation and model scripts are available publicly in an open source manner at http://www.covid-net.ml.

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
