# Peer review of "COVID-Net CXR-S: Deep Convolutional Neural Network for Severity Assessment of COVID-19 Cases from Chest X-ray Images"

_diagnostics, 2021, doi:10.3390/diagnostics12010025_

Round 1
Reviewer 1 Report
This paper proposed a deep learning framework namely COVID-Net CXR-S to predict the airspace severity of the Covid-19 patients using X-ray images. According to sensitivity and PPV, the proposed approach outperforms the state-of-art algorithms. Overall this is a well-written paper with technical sound results. The reviewer would suggest minor revisions based on the current version of the manuscript.
First, the architecture of the network is unclear in Section 2.1. The authors are encouraged to illustrated using a Table or Graph to present the internal architecture of the deep convolutional neural network. It would be valuable for others to reproduce the results.
Second, in Table 3, the specificity and AUC should also be provided and discussed.
Third, in Fig 3, the authors visualized the regions that contain factors lead to the decision while the radiologist confirmed these findings. However, it is still insufficient to demonstrate its power. The authors should invite the radiologists to blindly draw the regions of interest and compute the overlap ratio between the AI-identified region and radiologist-identified region. If it has a high overlap ratio, the results would be promising.
Last, please consider cite the following paper within the manuscript:
Zhong, Z., Kim, Y., Plichta, K., Allen, B. G., Zhou, L., Buatti, J., & Wu, X. (2019). Simultaneous cosegmentation of tumors in PET‐CT images using deep fully convolutional networks. Medical physics, 46(2), 619-633.
Li, H., Deng, J., Feng, P., Pu, C., Arachchige, D. D., & Cheng, Q. (2021). Short-term Nacelle Orientation Forecasting using Bilinear Transformation and ICEEMDAN Framework. Frontiers in Energy Research, 697.
Kong, W., Dong, Z. Y., Jia, Y., Hill, D. J., Xu, Y., & Zhang, Y. (2017). Short-term residential load forecasting based on LSTM recurrent neural network. IEEE Transactions on Smart Grid, 10(1), 841-851.
After revision, this paper has the strong potential for publication.
Author Response
This paper proposed a deep learning framework namely COVID-Net CXR-S to predict the airspace severity of the Covid-19 patients using X-ray images. According to sensitivity and PPV, the proposed approach outperforms the state-of-art algorithms. Overall this is a well-written paper with technical sound results. The reviewer would suggest minor revisions based on the current version of the manuscript.
We thank the reviewer for finding the paper to be well-written with technical sound results. We have made the minor revisions suggested by the reviewer in the revised manuscript.
First, the architecture of the network is unclear in Section 2.1. The authors are encouraged to illustrated using a Table or Graph to present the internal architecture of the deep convolutional neural network. It would be valuable for others to reproduce the results.
We thank the reviewer for the valuable comments. We have now clarified the architecture of the network by including an additional table detailing the internal network architecture in the revised manuscript.
Second, in Table 3, the specificity and AUC should also be provided and discussed.
We thank the reviewer for the valuable comments. We have now included these metrics in the revised manuscript.
Third, in Fig 3, the authors visualized the regions that contain factors lead to the decision while the radiologist confirmed these findings. However, it is still insufficient to demonstrate its power. The authors should invite the radiologists to blindly draw the regions of interest and compute the overlap ratio between the AI-identified region and radiologist-identified region. If it has a high overlap ratio, the results would be promising.
We thank the reviewer for the valuable comment. Such a study involving careful drawing of regions of interest by the radiologist would be interesting but would require a much longer, new study given limited availability of radiologists. We have now discussed this great suggestion as future work in the revised manuscript.
We have attached the revised paper to this message.
Reviewer 2 Report
I thank the Authors for this interesting manuscript and for exploring the difficult field of chest X-ray severity grading of COVID-19 disease.
I have a few comments:
- Please, re-state Line 37-38 "Given that severity assessment using CXR images can be quite challenging for front-line healthcare workers without expertise in radiology", as it seems to intend CXR are interpreted by non-radiologists.
- Were the 16,000 CXR images used for transfer learning only of COVID-19 patients? please expand.
- Which dataset was used for validation after training? please explain better.
- Line 135: "Radiologist validation": how many cases were selected for further review by board-certified radiologists? Were the two radiologists' evaluation blinded? A table showing correlation between COVID-NET CXR-S results and the two single radiologists' results could be helpful in showing the performances.
- Did the obtained CXR severity results match with the clinical severity of COVID-19 infection?
- How well did COVID-NET CXR-S manage to distinguish COVID-19 infections from other pathological entities?
- Reference 15 and 17: please change the author to "Radiological Society of North America".
Author Response
I thank the Authors for this interesting manuscript and for exploring the difficult field of chest X-ray severity grading of COVID-19 disease.
We thank the reviewer for the kind remark, and we have done our best to address the comments below.
I have a few comments:
- Please, re-state Line 37-38 "Given that severity assessment using CXR images can be quite challenging for front-line healthcare workers without expertise in radiology", as it seems to intend CXR are interpreted by non-radiologists.
- We thank the reviewer for the valuable comment, and have restated the line as " Given that severity assessment using CXR images can be quite challenging even for expert radiologists"
- Were the 16,000 CXR images used for transfer learning only of COVID-19 patients? please expand.
- Which dataset was used for validation after training? please explain better.
- Did the obtained CXR severity results match with the clinical severity of COVID-19 infection?
- How well did COVID-NET CXR-S manage to distinguish COVID-19 infections from other pathological entities?
- Reference 15 and 17: please change the author to "Radiological Society of North America".
- We thank the reviewer for the valuable comment and we have made this change in the revised manuscript. We have attached the revised paper version to this message.